# Glyoxalase 2: Towards a Broader View of the Second Player of the Glyoxalase System

**DOI:** 10.3390/antiox11112131

**Published:** 2022-10-28

**Authors:** Andrea Scirè, Laura Cianfruglia, Cristina Minnelli, Brenda Romaldi, Emiliano Laudadio, Roberta Galeazzi, Cinzia Antognelli, Tatiana Armeni

**Affiliations:** 1Department of Life and Environmental Sciences, Polytechnic University of Marche, 60131 Ancona, Italy; 2Department of Clinical Sciences, Polytechnic University of Marche, 60126 Ancona, Italy; 3Department of Science and Engineering of Materials, Environment and Urban Planning, Polytechnic University of Marche, 60131 Ancona, Italy; 4Department of Medicine and Surgery, University of Perugia, 06132 Perugia, Italy

**Keywords:** glyoxalase system, glutathione, methylglyoxal, post-translational modification (PTM), metallo beta-lactamase

## Abstract

Glyoxalase 2 is a mitochondrial and cytoplasmic protein belonging to the metallo-β-lactamase family encoded by the hydroxyacylglutathione hydrolase (HAGH) gene. This enzyme is the second enzyme of the glyoxalase system that is responsible for detoxification of the α-ketothaldehyde methylglyoxal in cells. The two enzymes glyoxalase 1 (Glo1) and glyoxalase 2 (Glo2) form the complete glyoxalase pathway, which utilizes glutathione as cofactor in eukaryotic cells. The importance of Glo2 is highlighted by its ubiquitous distribution in prokaryotic and eukaryotic organisms. Its function in the system has been well defined, but in recent years, additional roles are emerging, especially those related to oxidative stress. This review focuses on Glo2 by considering its genetics, molecular and structural properties, its involvement in post-translational modifications and its interaction with specific metabolic pathways. The purpose of this review is to focus attention on an enzyme that, from the most recent studies, appears to play a role in multiple regulatory pathways that may be important in certain diseases such as cancer or oxidative stress-related diseases.

## 1. Introduction

More than 100 years have passed since the discovery of the glyoxalase system, initially identified as an enzyme capable of converting methylglyoxal (MGO) to lactic acid [1,2]. Much later, Racker showed that it consists of two enzymes, glyoxalase 1 (Glo1) and glyoxalase 2 (Glo2), that catalyze the formation of lactic acid from MGO in the presence of a catalytic amount of reduced glutathione (GSH) [3]. However, it was not until 1954 that the end product of MGO metabolism by the glyoxalase system was suggested to be D-lactate and not L-lactate [4], which was confirmed by Mannervik and co-workers only in 1973 [5]. Glo1 catalyzes the formation of *S*-D-lactoylglutathione (SLG) from the hemithioacetal (MeCOCH(OH)-SG) formed spontaneously from MGO and GSH. Glo2 catalyzes the hydrolysis of SLG to D-lactic acid, regenerating the GSH consumed in the first reaction. The system, better known as the glyoxalase system, acts in parallel with the glycolytic pathway and serves mainly to eliminate MGO produced spontaneously from α-oxoaldehydes and in particular from dihydroxyacetone phosphate and glyceraldehyde 3-phosphate, both intermediates in glycolysis [6]. MGO is a reactive α-ketothaldehyde formed mainly by the degradation of triose-phosphates but also by the degradation of threonines, the metabolism of ketone bodies and the fragmentation of glycated proteins. Therefore, the glyoxalase system prevents the accumulation of reactive α-oxoaldehydes and represents an enzymatic defense system against glycation [7]. MGO has multiple direct and indirect cytotoxic effects on the cell [8,9,10]. High concentrations of MGO modify proteins and nucleic acids, forming advanced glycation end-products (AGEs). The reaction of MGO with proteins to form AGE residues is direct and rapid, causing important amounts of hydroimidazolones derived from arginine residues, and in lesser quantity, lysine-derived adducts. For DNA, the most reactive nucleotide susceptible to glycation by MGO is deoxyguanosine (dG). The major nucleotide AGE detected is the imidazopurinone derivative (dg-MGO), but another adduct, N2-carboxymethyl-deoxyguanosine (CMdg), has also been reported. This condition immediately induces apoptosis in the cell, and results in cell dysfunction in the long term [11,12]. Moreover, the possible role of MGO as a signaling molecule, through protein modification and gene expression modulation, is emerging [13,14]. The glyoxalase system is ubiquitous and is present in the cytosol of all animal cells, plants, bacteria, fungi and protists. In various organisms, two or more Glo2 enzymes have been found and have been localized in different cellular compartments such as mitochondria and apicoplasts, depending on the organism [15,16,17]. In plants, the activities of glyoxalases and antioxidant defense systems are coordinated in reducing reactive oxygen species (ROS) and the high amount of MGO that increases as a consequence of abiotic stress [18]. In various human parasites, the glyoxalase system has been studied as a potential drug target and especially as an antimalarial target, given the importance of MGO as a toxic electrophile, which must be removed [19,20]. In mammals, only a single gene encoding Glo2 has been identified. This gene gives rise to two distinct mRNA species; one is a mitochondrially targeted Glo2 and the other is the cytosolic form of Glo2 [21]. Structurally, Glo2 enzyme consists of two domains. The first domain folds into a four-layered β sandwich, a metallo-β-lactamase domain; the second domain is predominantly α-helical, forming the substrate-binding site. The active site contains a binuclear metal center with Zn^2+^ as a metal ion [22].

Other than its canonical role in the glyoxalase system, little is known about Glo2. In recent years, the assumption that the glyoxalase system is involved in some post-translational regulatory modifications (PTMs) of proteins is gaining increasing evidence. Our group showed that Glo2 is able to promote S-glutathionylation of certain target proteins using its natural substrate, *S*-D-lactoylglutathione (SLG). *S*-glutathionylation is the specific post-translational modification (PTM) of protein cysteine residues, which are reversibly oxidized to mixed protein disulfides through the addition of the tripeptide glutathione [23,24]. The study reported for the first time that this modification is possible through a specific interaction of the catalytic site of Glo2 with the target protein [25]. Another important PTM in which Glo2 seems to be involved is the *N*-acetylation of the ε-amino of lysine residues of mitochondrial proteins. *N*-acetylation of lysine residues can occur by a mitochondrial *N*-acetyltransferase enzyme [26], but it can also occur non-enzymatically [27]. It has been shown that non-enzymatic *N*-acetylation of lysine residues in mitochondrial proteins often occurs via reversible *S*-acetylation of a proximal thiol [28]. The acetyl groups for these reversible PTMs are provided by AcCoA and *S*-acetylglutathione. It has been seen that in the presence of Glo2 and GSH, *S*-acetylation on cysteine residues is removed, and consequently, *N*-acetylation of lysine residues is limited. The mechanism underlying this change has not been studied, and the authors hypothesized that the role of Glo2 could be to degrade *S*-acetylglutathione and thus limit cysteine *S*-acetylation [28]. Finally, the most recent PTM that indirectly involves Glo2 is the lactoylation of protein lysine residues. Additionally, in this case, Glo2 is not directly involved in residue modification, but instead performs a controlling function on SLG concentration by maintaining, under optimal conditions, low and tolerable levels of SLG such that massive lactoylation of protein lysine residues is prevented [29]. Even roles for Glo2 in cell signaling have been demonstrated. In cancer cells, it has been shown that the upregulation of Glo2 with a proposed p63/p73-Glo2 regulatory axis could play a tumor-promoting role, at least in some cancer types [30]. In particular, in prostate cancer, it has been shown that activation of the pyruvate kinase PKM2-mediated estrogen receptor alpha (ERalpha) axis leads to Glo2 up-regulation, and that Glo2 expression via a mechanism involves the p53–p21 axis evading the apoptotic process and stimulating proliferation [31]. The involvement of glyoxalases has also been demonstrated in secretory processes. By modifying their activity, they can regulate the endogenous concentration of their substrates, which may themselves or their derivatives act as regulatory molecules in specific metabolic pathways. In this context, the link between endogenous SLG concentration and microtubule polymerization has been proposed several times, which still requires in-depth study. The aim of this review is to bring together most of the older and newer studies on Glo2 in order to provide an essential overview that can form the basis for future promising studies on other aspects of this fascinating enzyme.

## 2. The Glyoxalase System

The glyoxalase system is present in the cytosol of all cells. It consists of two enzymes: a *S*-lactoylglutathione lyase as Glo1 (EC 4.4.1.5) and a hydroxyacylglutathione hydrolase known as Glo2 (EC 3.2.1.6), together with a catalytic amount of GSH [32,33]. The system is aimed at converting α-oxoaldehydes into the corresponding α-hydroxyacids quickly and efficiently. Typically, the major physiological α-oxoaldehyde removed is methylglyoxal that is converted to D-lactic acid via the SLG intermediate. In the first reaction, the hemithioacetal 2-hydroxyacylglutathione formed spontaneously from MGO and GSH becomes the substrate for Glo1 that catalyzes isomerization to SLG. In the second reaction, Glo2 hydrolyzes SLG to the final product D-lactic acid and simultaneously regenerates the GSH consumed in the first reaction (Figure 1) [34]. The major source of MGO formation is from the degradation of triose-phosphates during glycolysis, minor sources are from the catabolism of threonine and ketone bodies and the fragmentation of glycated proteins [35,36,37]. The glyoxalase system prevents the accumulation of these reactive oxaldehydes and thus suppresses oxaldehyde-mediated glycation reactions [7,38]. Increased MGO levels have been linked to age-related diseases and many other pathological conditions such as diabetes, obesity, cancer and Alzheimer’s disease [39]. To prevent these conditions, glyoxalase enzyme levels increase, enhancing detoxification of MGO and thus protecting cells from its deleterious effects.

Recently, some authors reported the presence of another glyoxalase member, named glyoxalase 3 (Glo3). Glo3 catalyzes the conversion of MGO into D-lactate in a single step without requiring any cofactor [40,41]. Glo3 was reported to be a member of DJ-1/Pfp1 superfamily and was first identified in *E. coli* [42]. 

## 3. Genetics and Molecular Properties of Glyoxalases 2

Glyoxalases are usually encoded by a single gene in microbial and eukaryotic genomes [21,44,45,46], while in plants and yeast, multiple genes are present [47,48]. For Glo2, the cytosolic and the mitochondrial forms of the enzyme are encoded by separate genes in yeast and higher plants and by a single gene in vertebrates [21]. For instance, in the rice genome, three Glo2 genes are present, while in the *Arabidopsis thaliana* genome, there are five [48,49,50]. A genome-wide study confirmed the presence of multiple glyoxalases in plants, with *Glycine max* genome possessing twelve Glo2 genes [47]. In *A. thaliana*, five different genes of Glo2 have been identified and three of these isoforms appears to be mitochondrial (GLX2-1, GLX2-4 and GLX2-5) [50]. The subcellular localization, molecular mechanism and functional role are currently unknown. These multiple forms of glyoxalases genes in plants probably indicate a tissue-specific MGO detoxification. Some genes encode inactive Glo2 forms, which are found in one of the three rice Glo2 and in two of the five *A. thaliana* Glo2 [48,50,51], forms that probably possess a different function than that of a thiolesterase. As an example, this functional diversification has been confirmed for *A. thaliana* glyoxalase 2-1, which is the isoform that is essential and produced by the plant during abiotic stress but is not necessary in normal growing conditions. *A. thaliana* glyoxalase 2-1 shows high similarity (88%) to the functionally active glyoxalase 2-5 protein, lacks Glo2 activity, but possesses β-lactamase activity. It could be an example of ongoing gene evolution where the duplication and functional divergence of an ancestral mitochondrial Glo2 gene have led to the emergence of β-lactamase activity, although plants do not produce β-lactams. A functional investigation of the role of *A. thaliana* glyoxalase 2-1 shows that its loss-of-function mutants and constitutively overexpressing plants resemble wild-type plants under normal growth conditions, whereas during abiotic stress mutations in *A. thaliana*, glyoxalase 2-1 inhibits plant growth and survival [52]. The second Glo2 gene from *A. thaliana*, glyoxalase 2-3, which also lacks canonical Glo2 activity, has been shown to possess sulfur dioxygenase activity and is known to be critical for seed development and conditions that involve high protein turnover [53]. In plants, adverse environmental conditions such as extreme temperatures, salinity, drought and heavy metal toxicity are critical factors that drastically reduce crop yields [54]. Under different abiotic stresses, MGO increased and became toxic for cellular components [55,56]. The accumulation of MGO resulted in the inhibition of germination and cell proliferation in a dose-dependent manner [57,58,59]. In *A. thaliana*, root elongation was significantly reduced due to 1 and 10 mM MGO, and chlorosis occurred at 10 mM [60]. Thus, in recent years, the glyoxalase system and Glo2 have been studied in relation to abiotic stress tolerance. In *B. campestris*, exposure of 150 mM NaCl increased Glo2 activity [61]. On the other hand, Hasanuzzaman and colleagues showed a decrease in Glo2 activity with increased ROS production under salinity stress in *Triticum aestivum*, *B. napus* and *O. sativa* seedlings [62,63,64]. Overexpression of Glo2 in rice and tobacco can lead to better tolerance to high MGO under salinity stress [51]. Rahman et al. showed that supplementation with manganese and calcium increases Glo2 activity and reinforces MGO detoxification in salt-affected seedlings [65,66]. Not only salt-induced stress increased Glo2 expression, but also heavy metals and abscisic acid (ABA) stress. In *Brassica juncea*, exposure to ZnCl_2_ increased Glo2 transcript levels [67]. Treatment with the xenobiotic compound, 2,4,6-trinitrotoluene (TNT) of *A. thaliana* seedlings root, resulted in an increase in Glo2 transcripts [68]. Moreover, genome-wide expression studies in rice and *A. thaliana* have shown a differential response of multigene family of glyoxalases during different growth and reproductive stages and in different tissues under various abiotic stresses [48]. Stress-responsive elements such as ethylene-responsive elements, abscisic acid-responsive element (ABRE), auxin-responsive element (AuxxRR-core) and heat shock element (HSE) have been identified in the promoter region of Glo2 family members in *A. thaliana* and soybean (*Glycine max*), suggesting that these genes could be regulated by hormonal and stress response pathways [47,52]. In addition to abiotic stressors, glyoxalase genes are highly induced by biotic stress conditions in bacteria, fungi, viruses, parasites and insects. Ghosh and Islam have reported biotic stress-responsive *cis* elements such as the fungal elicitor-responsive element (BOX-W1), wounding- and pathogen-responsive elements (W-box and WUN-motif), jasmonate elicitor-responsive element (JERE), methyl jasmonate-responsive elements (CGTCA box and TGACG motif), defense and stress-responsive element (TC-rich) and salicylic acid-responsive element (TCA) in the promoter region of Glo2 [47]. The regulatory mechanisms for the glyoxalase expression remain unclear, and further studies must be carried out to determine if glyoxalase genes might offer protection from pathogens. 

Regarding the yeast genome, two isoforms of glyoxalase 2 have been found: Glo2p and Glo4p proteins, with different subcellular localization. Glo2p is a cytosolic isoform, while Glo4p is present in the mitochondrial matrix [69]. The GLO4 gene has been identified first as a multicopy suppressor gene for a mutant yeast strain, which has a reduced efficiency of spore germination. A second gene, named GLO2, corresponds to a protein that has glyoxalase 2 activity. The amino acid sequences of the deduced proteins showed high similarities to the sequence of the human glyoxalase 2. Analyses with mutants lacking either one or both glyoxalase 2 genes showed that (i) the proteins are localized in different cellular compartments; (ii) the two glyoxalase II isoforms are differentially expressed depending on the carbon source used: glucose or glycerol; and (iii) to obtain an active Glo4p protein in *E. coli* through heterologous expression, the putative mitochondrial transit peptide at the N-terminus had to be removed [69]. Similarly, in *P. falciparum*, there are two Glo2 genes; one encodes for the cytosolic protein and the other codes for a protein localized in the apicoplast of the parasite [70]. These two active Glo2 isozymes both show a binuclear metal center [71]. The first report of a glyoxalase pathway in apicomplexan parasites was in *Plasmodium falciparum*-infected erythrocytes [72]. Given the relevance in the cellular detoxification of methylglyoxal, the glyoxalase system has been studied as a potential drug target in some human parasites, such as *Plasmodium falciparum*, *Leishmania* spp. and *Trypanosoma* spp. Differences in the glyoxalase pathways have been identified between human and parasites. Interestingly, the trypanosomatid *Trypanosoma brucei* is missing the Glo1 enzyme but has two genes encoding Glo2 enzymes; of the latter, only one showed glyoxalase 2 activity [73]. The life cycle of malaria parasites alternates between a vector (mosquito) and a human host stage each with their own specific proteome [74]. Regarding Glo2 expression during the life cycle stages of *P. falciparum*, tGlo2 was found to be expressed only in trophozoites and gametocytes; no information was available for cGlo2 [75]. Blood stages of the *P. falciparum* are the principal cause for the clinical manifestation of disease. Glucose uptake of red blood cells increased 75-fold upon infection [76]. This excessive glucose consumption allows for rapid endomitotic nuclear divisions and a drastic increase in parasitemia [77]. As a consequence of this high glucose metabolism, both malaria parasites and their host cells show elevated MGO production [72]. Therefore, the glyoxalase system has received a considerable amount of attention as a possible anti-malarial target and for its possible anti-trypanosomal activity [19,20]. In contrast to most eukaryotic organisms, the glyoxalase system of the trypanosomatids, including *Leishmania* spp. and *Trypanosoma* spp., uses reduced trypanothione (N1,N8-bis(glutathionyl)spermidine) (TSH), an alternative low-molecular-weight (LMW) thiol, as the preferred substrate [78,79,80]. Trypanosomatids differ from all other organisms in their ability to conjugate glutathione and a polyamine, spermidine, to form TSH. Together with the NADPH-dependent flavoprotein, trypanothione reductase (TR), the dithiol form of trypanothione, provides an intracellular reducing environment in these parasites, substituting glutathione and glutathione reductase found in the mammalian host. TSH and TR are involved in defense against damage by oxidants, some heavy metals, and possibly xenobiotics [81]. Due to its crucial role in the protection of parasites from oxidative stress [82,83] and heavy metal toxicity [84], enzymes involved in the trypanothione biosynthesis pathway are considered interesting candidates for drug development [85]. Trypanosomatid Glo2, similar to all other glyoxalases 2, contains a binuclear metal center [46,79]. The crystal structure of the Glo2 from *L. infantum* has revealed that Lys143, Arg249 and Lys252 residues involved in GSH binding have been substituted with Ile171 residue and two Phe residues, Phe219-266, in order to accommodate the positively charged thioesters of T[SH]2 or glutathionylspermidine [46]. Barata et al. produced a mutant enzyme able to hydrolyze both glutathione and trypanothione-derived thioesters by replacing Tyr291 and Cys294 with Arg and Lys [86]. Substrate specificities and significant differences in catalytic sites between trypasomatid and human Glo2 give hope that specific inhibitors are an achievable goal in drug development. 

Gram-positive bacteria include bacilli (e.g., *Bacillus subtilis*, *Bacillus anthracis*, *Bacillus cereus*, *Bacillus megaterium*, *Bacillus pumilis*), staphylococci (e.g., *Staphylococcus aureus*, *Staphylococcus saprophyticus*) and streptococci (*Streptococcus agalactiae*), which produce bacillithiol (BSH) as an alternative LMW thiol, which serves similar metabolic and redox cofactor functions as shown for GSH in eukaryotes [87,88,89,90]. In the past few years, some researchers have started to investigate the role of BSH in the detoxification of reactive oxidants and electrophiles, such as methylglyoxal [91] and fosfomycin [92], as well as the protection and redox regulation of protein functions by protein-*S*-bacillithiolation during oxidative stress [93,94]. Chandrangsu et al. showed that the BSH-dependent glyoxalase system confers protection against MGO primarily through cytoplasmic acidification resulting from the activation of the KhtSTU K+ efflux pump and secondarily by converting MGO to D-lactate. The activation of the KhtSTU K+ efflux pump is caused by the production of S-lactoyl-BSH, catalyzed from glyoxalase 1 in the conversion of the hemithioacetal adduct, while glyoxalase 2 converts S-lactoyl-BSH to D-lactate [95].

In vertebrates, the single gene encoding Glo2 produces two transcripts of nine and ten exons. The transcript derived from the nine exons encodes for mitochondrial Glo2 that has the start codon AUG in a previously uncharacterized upstream part of the mRNA. Also in this transcript is a downstream start codon that encodes for the cytosolic form. The 10-exon transcript only encodes for the cytosolic form because it has an in-frame termination codon. The double initiation by alternative AUG codons is conserved in all species [21]. The gene for Glo2 is conventionally referred to as HAGH (hydroxyacylglutahione hydrolase). The molecular mass of the cytosolic form of Glo2 is around 29 kDa, while that of the mitochondrial is approximately 34 kDa. The different molecular masses of the two isoforms are due to the absence of the first 48 amino acids in the cytosolic sequence. In fact, the amino-terminal extension of the mitochondrial isoform of Glo2 (mGlo2) contains a mitochondrial targeting sequence that has been identified in spinach [17], *A*. *thaliana* [50] and vertebrates such as mammals, birds and fish. In human HepG2 cells, it was seen, by confocal fluorescence microscopy, that the mitochondrial form of Glo2 is present in the mitochondrial matrix [21]. In rat liver mitochondria, two separate pools of mGlo2 (purified by affinity chromatography) were found: one in the intermembrane space and the other in the mitochondrial matrix. From both crude and purified preparations, polyacrylamide gel-electrophoresis resolved multiple forms of Glo2, two from the intermembrane space and five from the matrix [96]. The presence of mGlo2 was also demonstrated in the bovine liver extracts, which accounted for about 10% of the total Glo2 activity of the whole homogenate. Electrophoresis and isoelectric focusing of the crude mitochondrial extract or purified mGlo2 resolved the enzyme activity into five isoforms (pl 6.3, 6.7, 7.1, 7.7 and 7.9). Since the bovine liver cytosol showed only one Glo2 isoform (pl 7.5), it was assumed that at least four isoforms were exclusively mitochondrial [97]. Our study also showed differences between cytosolic and mitochondrial Glo2. Indeed, in this study, it was shown that the activity of cytosolic Glo2 was inhibited by contact with liposomes formed from negatively charged phospholipids, whereas no inhibition on enzymatic activity was detected on mGlo2 [98]. Considering that a single gene encodes Glo2 (HAGH) and that there are two mRNA transcripts, it is possible to assume that there are other post-transcriptional rearrangements that give rise to different mitochondrial isoforms. The mGlo2 within the mitochondria has SLG as its substrate of choice but also hydrolyzes other acyl-GSH derivatives such as *S*-acetyl-GSH and S-succinyl-GSH. One of our studies showed that SLG can enter mitochondria and provide the substrate for mGlo2, and it was hypothesized that this could be an alternative GSH supply route for the mitochondria and ATP production through the oxidation of D-lactate to pyruvate by the enzyme D-lactate dehydrogenase [99]. It was pointed out that it is unlikely that SLG is a substrate for GSH uptake in mitochondria since SLG levels are low, usually less than 1% of GSH, and a role as an acceptor of acetyl groups was proposed [34,100]. In light of the new findings concerning the involvement of Glo2 in PTMs, there is a good chance that mGlo2 can be involved in the post-translational regulation of proteins (see paragraph 5), and in particular that GSH derived from the hydrolysis of SLG can be utilized by mGlo2 for the *S*-glutathionylation of specific target proteins.

Recently, the presence of Glo2 protein into the nuclei of human prostate cancer cells but not in normal cells has been reported [101]. Further investigation will be required in order to identify as best as possible the nuclear localization of Glo2 and its possible role(s). The amino acid sequences of Glo2 proteins from different species revealed two different domains (Figure 2). One is the metallo-β-lactamase domain (present in all members of the metallo-β-lactamase superfamily and required for the catalytic activity), and the other is a hydroxyacylglutathione hydrolase C-terminus (HAGH-C) domain that forms the substrate-binding site (usually present at the C-terminus of Glo2 proteins) [22]. The length of these domains varies among different species, but the overall domain architecture of Glo2 proteins remains the same (Figure 2).

## 4. Enzyme Activity, Structural Features and Catalytic Mechanism

For their activity, Glo2 enzymes require a binuclear metallosite, generally two divalent zinc ions [44,45,49]. However, this is not an absolute requirement, as different metals such as iron, manganese or zinc have been used [50,102]. For instance, Glo2 enzymes of prokaryotes and eukaryotes, including plants, possess a binuclear metal binding center, binding iron and zinc [22]. Glo2 enzymes from rice [103] and *A. thaliana* [50,104] analogously have been found to possess a Zn/Fe binuclear center, similar to their homologs from *Leishmania infantum* [46] and humans [44]. In particular, *A. thaliana* cytoplasmic GLX2 (GLX2-2) can bind Zn^2+^, Fe^2+^ and Mn^2+^ [22,105,106], while the human enzyme was reported to bind two Zn^2+^ ions [44]. The calculated kinetics parameters follow Michaelis–Menten kinetics and consist of a *k*_cat_ value of 2.8 × 10^2^ s^−1^ and a *k*_cat_/*K*_m_ value of 8.8 × 10^5^ M^−1^ s^−1^ for the *S*-D-lactoylglutathione substrate. Among the thiol esters of glutathione that have been tested, *S*-D-lactoylglutathione showed the maximum velocity and represents the main substrate of Glo2. The catalytic efficiency of Glo2 was similar in humans, plants and yeast [69,107,108]. The enzymatic activity was found salt- and pH-sensitive. Urscher and Deponte demonstrated that *P. falciparum* Glo2 activity decreases with high NaCl concentration due to an increased Km value and a decreased kcat value [109]. This result was in agreement with the experiments on human Glo2 [110]. For the hydrolysis of *S*-D-lactoylglutathione, the presence of a group with a basic pKa value was detected for rat, human and *P. falciparum* Glo2 [109,111,112] and could be influenced by acid–base catalysis at the binuclear metal center. Several competitive glutathione-based inhibitors were synthesized and studied by computational analysis [113,114,115]. Little is known about Glo2 inhibition mechanism, as only a few studies reported different results on human, *A. thaliana* and *P. falciparum* Glo2 [106,109,110]. Urscher and Deponte proposed a Theorell–Chance inhibition pattern, a special case of an ordered Bi–Bi mechanism in which the concentrations of the enzyme–substrate and the enzyme–product complexes (EAB and EPQ) are essentially close to zero. According to this mechanism, the first product (P) is immediately released, whereas the second product (Q) dissociates more slowly. In agreement with this theory, substrate binding becomes a rate-limiting step [109]. A better understanding of Glo2 inhibition mechanism could lead to the development of new anti-cancer and anti-protozoan drugs.

Three-dimensional protein structures for a variety of Glo2 representatives have been reported and include Glo2 from *H. sapiens*, *A. thaliana*, *Leischmania infantum* and *Salmonella* Typhimurium [44,50,102,116]. The crystal structure of human Glo2 was first reported in 1999 [44]. The protein, formed by 308 amino acids, consists of two structural domains. The N-terminal domain, including residues 1–173, is structurally similar to the whole structure of a generic metallo-β-lactamase (Figure 3a) [117]. The second domain is situated at one edge of the first and consists of residues 174–260 folded into five α helices. The active site contains an Fe(II)Zn(II) center. Specifically, five histidines (His 54, His 56, His 59, His 110, His 173) and two aspartate residues (Asp58, Asp134) interact directly with two zinc ions [44]. In more, Zn1 is coordinated by His54, His56 and His110, while Zn2 is coordinated by His59, His173, Asp58 and Asp134 (Figure 3b) [118]. The active site is characterized by a αβ/βα fold along with the conserved THxHxDH motif, able to bind up to two metal ions, as is typical for a β-lactamase [117].

In addition to the coordination of metal ions, the active site represents the pocket in which the substrates that are recognized and processed by Glo2 can react. The structure of human Glo2 has also been characterized by studying the unfolding equilibrium and refolding: the denaturation at equilibrium of wild-type Glo2 is multiphasic, suggesting that the native structure of Glo2 is made by different regions of varying structural stability. At an intermediate denaturant concentration (1.2 M guanidine), a molten globule state is attained. Only in the presence of Zn(II) ions, the denatured wild-type enzyme reactivates. The results show that Zn(II) is essential for the maintenance of the native structure of Glo2 and that its binding to the apoenzyme occurs during an essential step of refolding [119]. The action mechanism in the active site was also studied theoretically for human Glo2 and it is reported in Figure 4. After the binding of the substrate, the first step is a nucleophilic attack of the oxygen to the carbonyl carbon, which forms a tetrahedral intermediate. The second step involves breaking the C–S bond and the concomitant formation of the double bond between C and O. The third step involves the regeneration of the active site, and so the products are released and a new water molecule enters to form a bridging hydroxide. It is clear that the two metal ions each have a specific role: while Zn2 stabilizes the formation of the tetrahedral intermediate in the first step, Zn1 helps to break the C–S bond in the second step by interacting with the negatively charged sulfur [120].

The proposed mechanism is also in agreement with experimental findings suggesting that even if Glo2 contains an Fe(II)Zn(II) center in vivo, the catalytic activity is due to Zn(II) in the Zn1 site, that, as reported above, is directly involved in C–S bond breaking, while Zn2 only stabilizes the tetrahedral intermediate. This role can also be assumed by other metals such as Fe(II) or Ni(II) in Glo2 homologs of other species. The same mechanism was previously reported for *A*. *thaliana* Glo2, where it was reported that the metal ions could be zinc and iron [106]. The effect of different metal ions on the activity of Glo2 for different species has been the subject of some studies [22,67,104,106,121]. These indicate that together with the structural observations, generally, Glo2 must be able to work in different conditions for the plants, because they cannot move, so the enzyme must be active with the different metal ions available where the plant grows. As an example, it has been reported that mitochondrial Glo2 of *B. juncea* is upregulated by salinity and heavy metal stress [67]. It is interesting to note that for Glo2 of *S*. Typhimurium [122], a metal-selective product inhibition was observed, indicating that the iron variant is able to form a stable enzyme–product complex, while the manganese derivative is not. The different metal forms of Glo2 during *Salmonella* infection could be exploited as a mechanism to regulate enzyme activity.

Besides these studies and findings, aiming to elucidate, from a structural point of view, the role of Glo2 in the mechanism of *S*-glutathionylation, a reliable computational protocol consisting of a protein–protein docking approach followed by atomistic Molecular Dynamics (MD) simulations was set up to predict the macromolecular association of Glo2 both in the presence or absence of GSH, with proteins known to be in vitro *S*-glutathionylated, such as actin, malate dehydrogenase (MDH) and glyceraldehyde-3-phosphate dehydrogenase (GAPDH) [123]. Computational protocol and MD simulations confirmed the previous experimental and in silico evidence, where it was shown that Glo2 has a high affinity of interaction with actin and MDH through its active site, which is much higher in presence of GSH, thus suggesting a possible role in the process, while no interaction was shown with GAPDH [25].

## 5. Glo2 Role in Post-Translational Modifications

The enzyme Glo2 directly and indirectly participates in post-translational modifications (PTMs) of lysine and cysteine residues by *S*-glutathionylation, *N*-lactoylation and *S*-acetylation (Figure 5).

Glo2 catalyzes the hydrolysis of SLG, forming D-lactate and releasing glutathione. The energy released by this process is comparable to that released in the cleavage of an ATP molecule to ADP, so it was necessary to ask where this energy converged during the hydrolysis of SLG. In 2004, our research group hypothesized that Glo2 could play an additional role to that of a thiolesterase, using glutathione in the SLG form to specifically glutathionylate certain redox-sensitive proteins, thus paving the way for the hypothesis on the involvement of the glyoxalase system in post-translational regulatory modifications of proteins. *S*-glutathionylation is a PTM that can regulate the activity of multiple redox-sensitive proteins, and its modifications are present in several oxidative stress-related diseases [124,125,126].

Our studies have shown the ability of Glo2 to specifically *S*-glutathionylate two different proteins allocated to two different cellular compartments: actin and malate dehydrogenase (MDH). In these in vitro experiments, it was shown that actin was specifically *S*-glutathionylated at residue Cys374 in the presence of Glo2 and its substrate SLG, and that this event was dose-dependent. In addition, in the presence of *S*-glutathionylation, the hydrolase activity of Glo2 was lowered, demonstrating that when the enzyme is engaged in *S*-glutathionylating proteins, its enzymatic activity slows down, suggesting an allosteric binding site to proteins undergoing *S*-glutathionylation and a direct transfer of GSH in the form of GS^−^* from the enzyme’s active site to the cysteine residues in the target protein (see Figure 5) [25,123]. In *S*-glutathionylation experiments conducted in vitro, GSSG is usually used at high concentrations as a positive control and results in non-specific *S*-glutathionylation by forming disulfide bridges with all the sulfhydryl groups of the cysteine residues in proteins [127]. In our experiments, *S*-glutathionylation is triggered by Glo2. Cysteine residues on proteins are its targets, and the substrate is at micromolar concentrations, although it is difficult to determine whether these concentrations can be reached inside the cell. Glo2 also exhibited target specificity towards the enzyme MDH (besides actin), and no post-translational *S*-glutathionylation modification on GAPDH was detected, although this is an enzyme regulated by *S*-glutathionylation and is sensitive when exposed to high concentrations of GSSG, demonstrating Glo2’s target specificity only for selected proteins. The specificity of Glo2 towards actin is interesting because actin is fundamental for the cell shape and function by playing mechanical, organizational and signaling roles. Reversible *S*-glutathionylation regulates actin polymerization, showing an inhibition when Cys374 is *S*-glutathionylated [128]. For example, oxidative modifications to actin that lead to polymerization and/or depolymerization influence the migration, proliferation and contraction of vascular cells, thus revealing an important role in vascular diseases [129]. Indeed, actin dynamics are crucial for the regulation of endothelial barrier stability and vascular permeability [130], and agents that disrupt the cortical actin rim, including oxidants and oxidative stress, increase endothelial permeability [131]. Remodeling of actin is frequently mediated by rapid assembly and disassembly of actin filaments, a process that depends on many actin-binding and actin-regulating proteins, as well as on the ratio between filamentous and globular actin (F-actin and G-actin, respectively) [132]. Dalle Donne et al. showed in vitro the *S*-glutathionylation of actin, which was achieved with a high concentration of non-physiological GSSG [133,134] and showed that *S*-glutathionylation of Cys374 in actin reduces the ability of G-actin to polymerize in F-actin [134,135]. Other studies showed that the steady-state rate for non-glutathionylated actin polymerization was at least 5.6-fold faster than that of the glutathionylated actin [128]. The degree of actin polymerization modified through *S*-glutathionylation is shown in many studies involving both cytoskeletal rearrangements during regular cellular functions and in a wide variety of pathological conditions [136,137,138,139]. *S*-glutathionylation of MDH was even more interesting because it immediately showed the correlation with enzyme activity. MDH was selected for the *S*-glutathionylation tests because during the chromatographic purification of Glo2 from rat liver, it was bound to the enzyme and then identified by sequencing. In this study, in vitro incubation with MDH and GSSG as positive control resulted in strong *S*-glutathionylation, although the protein remained in monomeric form. Incubation with Glo2 and its substrate SLG showed *S*-glutathionylation of the protein and was also able to promote monomeric MDH to the tetrameric conformation. The latter result is significant because MDH has eight cysteine residues that can be *S*-glutathionylated differently, and selective modification of these residues is able to change the conformation of the active site and, consequently, to modify the level of activity of the enzyme [140,141]. It is therefore possible to assert that Glo2 is able to catalyze *S*-glutathionylation on specific cysteines of MDH, promoting its tetramerization state and thus influencing its metabolic activity, whereas GSSG leads to non-specific *S*-glutathionylation on all exposed cysteines, which seems to be a non-specific protective role against oxidation of cysteine residues in a redox environment [142]. It was indeed seen that MDH activity increased significantly in the presence of Glo2 plus SLG, probably caused by the tetramerization of the enzyme, whose state correlates with its kinetic activity [143]. On the other hand, the enzymatic activity of Glo2 decreased, highlighting the involvement of the active site during *S*-glutathionylation, also confirmed by the protein–protein docking analysis study [123]. Glo2 is also involved in another important post-translational modification involving the *S*-acetylation of cysteine thiol residues (Figure 5). Indeed, it has been shown that non-enzymatic lysine *N*-acetylation by AcCoA, in mitochondrial proteins, was greatly enhanced by an S-acetylated thiol intermediate proximal to the lysine, which was followed by an SN transfer of the acetyl moiety to the neighboring lysine [28]. It has been shown that Glo2 can limit the AcCoA-dependent modification on cysteine residues (*S*-acetylation) and consequently limit the lysine *N*-acetylation. This type of regulation is important at the mitochondrial level since AcCoA not only provides acetyl groups to the citric acid cycle, but it also provides acetyl groups for the *N*-acetylation of lysine residues on proteins that are involved in a large number of regulatory processes [27,144,145]. The importance of *N*-acetylation is demonstrated by the existence of a mitochondrial deacetylase, sirtuin 3, which regulates this post-translational modification on proteins. The mitochondrial deacetylase Sirt3 is important in a range of degenerative diseases including ageing, cancer and diabetes. In addition to Sirt3, carnitine acetyltransferase (CrAT) has been shown to buffer AcCoA concentration by storing excess AcCoA as non-reactive O-acetylcarnitine, thus limiting thioester concentration and consequently acetylation of cysteine and lysine [146]. James et al. identified mitochondrial Glo2 as another system that is able to limit CoA-derived modifications. In this study, the authors showed that in the presence of Glo2, *S*-acetylation is not present, which goes to limit *N*-acetylation of proximal lysine residues. It is possible that Glo2, in the mitochondrial matrix, is able to degrade *S*-acetylglutathione, shifting the balance away from *S*-acetylation of cysteine and thus limiting acetylation of cysteine and lysine residues, although the mechanism underlying this change has not yet been studied [28]. GSH at physiological concentrations is ineffective, because the reaction product is *S*-acetylglutathione, which could simply re-acetylate proteins [147]. Glo2 is of advantage because it increases the rate of hydrolysis of a number of acylglutathione species without affecting the corresponding AcCoA [96].

Another important post-translational modification in which Glo2 is indirectly involved is the lactoylation of lysines, which consists of a non-enzymatic acyl transfer of the SLG moiety lactate to the protein lysine residues (Figure 5). Gaffney et al. identify the non-enzymatic acyl transfer of the lactate moiety from SLG to the lysine residues of proteins, generating a lactoyl-lys modification on proteins. In this study, it was reported that cells lacking Glo2 show high levels of SLG and consequently also a marked increase in the lactoylation of lysines. The target molecules for this PTM were mainly glycolytic enzymes [29]. In this study, protein lactoylation is detected using 1 mM SLG. It should be noted that this is an extremely supraphysiological concentration of SLG, and therefore, the system provides limited information on the function of SLG physiologically in situ [99]. From this point of view, we can affirm that Glo2 performs an important cytoprotective function by maintaining low and tolerable levels of SLG to prevent protein lactoylation. In fact, with an eventual SLG accumulation, there could be a significant transfer of the lactoyl moiety to cysteine forming *S*-D-lactoylcysteine that rearranges to *N*-D-lactoylcysteine [148]. Thus, in the case of a high MGO production due to accelerated glycolytic flux, a transient increase in the amount of SLG could potentially occur, leading to protein lactoylation. This alteration, once it occurs, regulates the speed of glycolysis by inhibiting the activity of some glycolytic enzymes undergoing lysine lactoylation [29]. Glo2 is the only known cytoplasmic enzyme in mammalian cells that hydrolyzes SLG, so Glo2’s enzymatic activity becomes a particularly important checkpoint for PTMs that regulate key metabolic pathways such as glycolysis. Zhang et al., in 2019, showed lactate-derived histone lysine lactylation on 28 sites on core histones in human and mouse cells, highlighting an important new epigenetic modification on chromatin [149]. These authors consider lactyl-CoA to be the substrate of choice for lysine lactoylation, and on this question, an interesting debate has opened with other scientists. In fact, Khadka et al. have expressed some criticism of the presence of lactyl-CoA within the cell, as this compound has not yet been quantified in the Human Matabolome Database (HMDB0002346), so even if present, it must be in undetectable quantities. Instead, these authors focus their attention on SLG as a substrate for the nucleophilic lactyl substitution of lysine residues on histones [150]. On the other hand, the non-enzymatic transfer of MGO and SLG adducts to lysine had already been demonstrated by Gaffney et al., as described above. Kulkarni and Brookes also express some concerns regarding the origin of the substrate for this new PTM. These authors similarly propose SLG as a probable metabolite for lactyl-lysine modification, although the concentration of the latter substrate is also at the limit of detection. These authors importantly show that L-lactate inhibits Glo2, and they suppose this inhibition may lead to SLG accumulation. This mechanism could provide a link between the rate of glycolysis and lysine lactylation [151]. On the other hand, MGO is also strongly linked to the glycolysis, with its ability to rapidly inactivate certain glycolytic enzymes such as glyceraldehyde 3-phosphate dehydrogenase and fructose bisphosphate aldolase through the involvement of the arginine residue located at the anionic binding site of the phosphate group of the substrate [152]. It is possible that the two glyoxalase pathway substrates act via PMTs on the same targets to achieve the same regulatory effect on certain pathways, as observed for glycolysis. We must consider that the concentration of metabolic pathway substrates is generally regulated by enzyme activity, so Glo1 and Glo2 enzymes can be considered as regulators of certain intracellular metabolic activities. Thus, for example, amplification of Glo1 enzyme activity without an increase in Glo2 activity can lead to an increase in LactoylLys generation and consequent potential slowing of glycolytic flux and thus also the proliferative potential of cells. Indeed, it has been observed that prostate cancer cells have high levels of Glo2 activity [101], which does not allow SLG to accumulate at the cellular level. On the other hand, it has been seen in many cell lines that Glo2 always has a higher activity than in normal cells and also appears to be an anti-apoptotic factor [30]. In this scenario, it is possible to assume that Glo2 is a central enzyme in the control of three important post-translational modifications (*S*-glutathionylation, *N*-acetylation and lysine lactoylation) involved in the cell metabolic regulation.

## 6. Glo2 Metabolic Interactions

### 6.1. Glyoxalase 2 and Signaling Pathways

The study of the signaling pathways regulating Glo2 expression as well as those driven by Glo2 to control specific biological responses dates back to 2006 when Xu and colleagues [30] demonstrated, for the first time, in breast cancer MCF7 cell lines, that Glo2 can be up-regulated by the p53-related genes p63 and p73 through a specific responsive element located in the intron 1 of Glo2 gene. They also found that, upon overexpression, the cytosolic (cGlo2), but not the mitochondrial Glo2 (mGlo2), inhibited the apoptosis of MGO-induced MCF7 breast cancer cells. Similarly, cGlo2 knockdown enhanced it. Since Glo2 and Glo1 are frequently overexpressed in cancer as a strategy to avoid accumulation of high levels of cytotoxic MGO, forming in local hypoxic malignant environment by increasing the level of glycolysis [153], and p63 and p73 [154] are overexpressed in some types of cancers [155], where they might have a tumor-promoting role [154], the authors suggested that the p63/p73–Glo2 axis might have been a novel pathway in human carcinogenesis, at least in some cases. Between 2006 and 2007, we reported that Glo2 was differently regulated by estrogens in breast cancer cell lines [156], while it was regulated by the steroid hormone testosterone and estradiol assumable as part of an intracellular response mechanism to the androgen-induced oxidative stress or to the presence of androgen response elements (ARE) in Glo2 promoter, as predicted by bioinformatic analysis [157]. More recently, we demonstrated that Glo2 is regulated also by PTEN/PI3K/AKT/mTOR signaling via pyruvate kinase (PK)M2-mediated estrogen receptor alpha (ERα) activation as a novel mechanism to drive prostate cancer (PCa) progression [31]. In particular, in aggressive models of PCa, we found that PTEN loss induces PTEN/PI3K/AKT/mTOR activation that in turn activates the p-PKM2/Erα axis, leading to Glo2 up-regulation, associated with increased cell survival, proliferation, migration and invasion (Figure 6).

Finally, Glo2 expression in PCa is under androgen receptor (AR) control to stimulate cell proliferation and elude apoptosis through a mechanism involving the p53–p21 axis [101]. In the same year, Dafre and colleagues published results pointing out that Glo2 degradation was induced by autophagy upon MGO exposure [158]. More recently, we showed that mitochondrial Glo2 can be upregulated by the bioactive compound oleuropein (OP), an olive-derived polyphenol with an array of pharmacological properties, including anti-inflammatory and antioxidant effects [159], to induce apoptosis of human non-small-cell lung cancer (NSCLC) A549 cells. Specifically, OP induces superoxide dismutase 2 (SOD2) upregulation with the consequent scavenging of superoxide anion O_2_^•−^. The depletion of O_2_^•−^ inhibits the Akt signaling pathway that, in turn, induces the upregulation of mitochondrial Glo2 expression. mGlo2 interacts with the proapoptotic protein Bax, activating apoptosis through the intrinsic pathway (cytochrome c (Cyt c) release from the mitochondrion, activation of Apaf-1, and eventually caspase-3 activation) (Figure 7). 

Apart from these studies, additional Glo2 regulation by other signaling pathways has not been reported, which encourages research to further explore this interesting field.

### 6.2. Glyoxalase 2 and Microtubules Interaction

In a number of non-recent studies, experiments were conducted showing the involvement of Glo2 in microtubule assembly [160]. In one of these studies, the authors prepared several competitive Glo2 inhibitors, including the highly specific and potent Glo2 inhibitor N,S-bisfluorenylmethoxy-carbonyl glutathione (DiFMOC-G). Treating rat chloroleukaemia cells in culture showed growth inhibition, which the authors hypothesize may be related to inhibition of microtubule polymerization. In order to verify this hypothesis, they treated embryotic neural cells with this Glo2 inhibitor. Since embryonic neuronal cells undergo major extensions to form axons and neurites within a few days of being cultured, requiring extensive synthesis of the microtubule cytoskeleton, the experiments were carried out both at 2 days after being cultured and at 15 days where the neuronal structures are already formed. Only cells that had been in culture for two days showed structural changes after they received DiFMOC-G treatment. Already after 24 h, the cultured neurons showed profound effects on axons and dendrites that were contracted or disappeared, while 47–72 h after treatment, the cells were completely globular in shape. In the same study, the authors show that Glo2 is associated with tubulin on calf brain preparations and that it remains associated after three cycles of aggregation/disaggregation. Nevertheless, when Glo2 was associated with tubulin, the inhibitors used resulted in substrates for Glo2 (comparable to SLG), and the results on polymerization were fairly equivocal [161]. On the other hand, Clelland and Thornelly demonstrated that SLG produced a small but significant increase in GTP-promoted assembly of microtubule proteins from the third loop of the porcine brain in a cell-free system. In this system, the presence of Glo2 inhibited the GTP-dependent potentiation of microtubule assembly [162]. Other experiments by other authors also show discordant and/or equivocal results such that none of them clarify the role of Glo2 on microtubule activity [163]. At the time when the above-mentioned research was conducted, the importance of post-translational modifications was still unknown. A relatively more recent study shows that microtubules can be *S*-glutathionylated and that functional thiol groups play a critical role in microtubule polymerization. Specifically, a correlation between protein *S*-glutathionylation and microtubule depolymerization was shown [164]. Taking all the information we have on Glo2, we can justify Clelland and Thornalley’s results because it is possible that Glo2 is involved in the *S*-glutathionylation of microtubules (our manuscript in submission), so in their experimental system, the presence of SLG and Glo2 leads to *S*-glutathionylation of microtubules and inhibition of polymerization. In Norton et al.’s study, inhibitors of Glo2’s thioesterase activity were used, and it has never been investigated whether these substrates can also inhibit Glo2’s *S*-glutathionylation activity. It is therefore possible to assume two different possibilities: the first is that the same inhibitors were used by Glo2 to glutathionylate the microtubule proteins and thus block polymerization during the first few days of culture in neuronal cells; the second is that these inhibitors induced oxidative stress with GSSG formation, which likewise leads to *S*-glutathionylation of the proteins and blocking of microtubule polymerization, as shown by Carletti et al. [165]. An important study by Gillespie showed the enhancement of anti-IgE-induced histamine release by SLG and contextually an inhibition of histamine release after treating cells with an inhibitor (PAL6-25-1) that induced a decrease in endogenous SLG. This study demonstrated the involvement of SLG as a modulator of anti-IgE-induced histamine release and thus the effect of this compound on the secretory system [166]. The same author shows that concanavilin A causes histamine release from basophilis through interaction with IgE and simultaneously activates both Glo1 and Glo2 in a dose-dependent manner in lymphocytes and polymorphonuclear leukocytes [167]. Since a role of SLG in microtubule assembly has been shown in vitro [168] and histamine release is a secretory process dependent on microtubule assembly, Gillespie hypothesized that the effect of SLG on histamine release is indeed related to the promotion of microtubule assembly [166]. In light of the new findings on the involvement of Glo2 in PTM, we can say that the regulation of enzymes in the glyoxalase system can serve to increase or decrease the two substrates of each enzyme that are important in the regulation of cell signaling. In particular, increasing Glo1 at the simultaneous decrease in the hydrolase activity of Glo2 may not lead to an accumulation of endogenous SLG but to a higher non-enzymatic activity of Glo2. Specifically, it has been observed that when Glo2 is activated for the *S*-glutathionylation of specific target proteins, its enzymatic activity decreases, so it is possible to suggest that the decrease in Glo2 activity, when occurring at the same time as the increase in Glo1, can lead to a greater substrate for *S*-glutathionylation by Glo2. On the other hand, this situation is not always the case, because sometimes an increase in both enzymes has been demonstrated, e.g., during secretion processes, so in order to further investigate the regulation of the glyoxalase system, it would be appropriate to measure the ratio between the respective enzyme activities of the two enzymes. In conclusion, it can be claimed that there is probably an interaction between Glo2 and microtubules in terms of PTM, but more extensive studies must be carried out in order to investigate its possible role in microtubule polymerization and other regulative PTMs.

## 7. Conclusions

Although the glyoxalase system has been studied for more than a hundred years, it still has many unknown aspects. In particular, the enzymatic regulation of the two enzymes that make up the MGO pathway may be of fundamental importance in cell regulation under certain conditions, such as in cancer or in pathological conditions that induce oxidative stress. Glo2 is widely distributed in nature, and by using fluorescent antibodies, it is possible to detect with confocal microscopy a large presence of this enzyme in the cytosol, a presence that overlaps with the cytoskeleton, indicating a possible fundamental role in cells, although Glo2 knockout does not imply any detectable change on cultured cells [169]. This latter finding certainly indicates a regulatory role only under specific cell conditions; for example, in plants, Glo2 plays a key role during abiotic stress, although under normal conditions, this enzyme does not appear to be essential for plant survival (see above for references). The same conditions could occur in animal cells; during conditions of oxidative stress, it is important that Glo2 is activated, whereas under conditions of oxidoreductive homeostasis, it can be silenced without compromising cell survival. On the other hand, we must consider that it is an enzyme that hydrolyzes SLG to release, in addition to D-lactate, glutathione, so it is part of the pool of enzymes that the cell can use during conditions of redox imbalance. Glutathione can either be released or it can be utilized by Glo2 to bring about *S*-glutathionylation on target proteins, and this aspect indicates a regulatory role for the enzyme that can be more activated for this function under specific pathological conditions. It is interesting to note that in mammalian cells, there is a cytosolic Glo2 and a mitochondrial Glo2 that naturally have different interactions and roles. It has been shown that SLG can enter mitochondria and provide the substrate for mitochondrial Glo2, and from the studies described above, it is clear that mitochondrial Glo2 can play multiple roles in regulating processes such as apoptosis. Indeed, it has been shown that Glo2 can interact with mitochondrial Bax, can limit the *S*-acetylation of lysine residues and can induce *S*-glutathionylation. The role of a PTM-specific protein opens up great regulatory possibilities at both the cytosolic and mitochondrial levels that are worth investigating and studying. Many studies on Glo2 have not continued and have never been investigated in depth mainly because the substrate is easily hydrolyzed, the enzyme is sensitive to external redox conditions and the anti-Glo2 antibody has been available for relatively few years. For example, one of our studies showed that there were specific interactions between cytosolic Glo2 and liposomes consisting of negatively charged phospholipids that influenced and inhibited its hydrolase activity, whereas this inhibition on mitochondrial Glo2 was not detected. The data were discussed in relation to a possible regulation of SLG at the cytosolic level such that it was allowed to accumulate and be directed to the mitochondria, which could utilize it via mGlo2, which was not inhibited [98]. Other studies have not been further investigated, such as those concerning the interaction with the cytoskeleton and the positioning of Glo2 in cellular compartments. To date, these studies open up new research possibilities in the field of the glyoxalase system, the knowledge of which can be applied for a basic understanding of the cell but also to the study of pathological conditions.

## Figures and Tables

**Figure 1 antioxidants-11-02131-f001:**
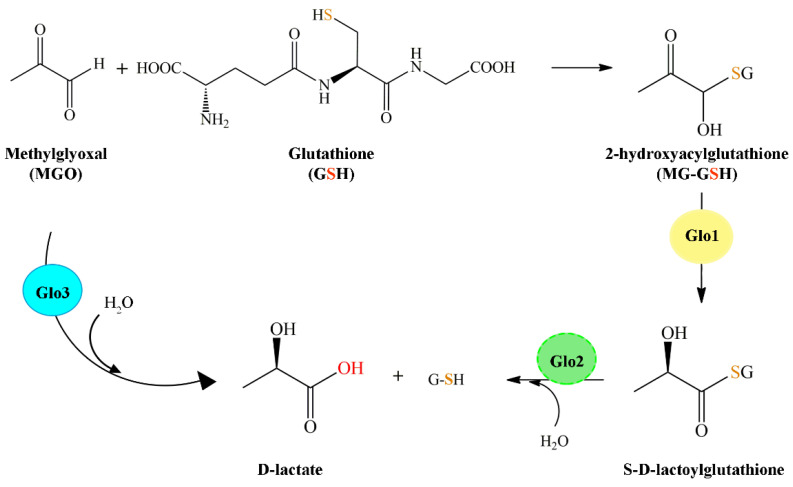
Conversion of methylglyoxal to D-lactic acid by glyoxalases. The GSH-dependent system is a two-enzyme pathway involving Glo1 and Glo2 enzymes. Glo1 catalyzes the formation of *S*-D-lactoylglutathione from the non-enzymatically formed hemimercaptal adduct of MGO with GSH, 2-hydroxyacylglutathione (MG-GSH). Glo2 catalyzes the hydrolysis of *S*-D-lactoylglutathione to D-lactic acid and GSH [43]. The GSH independent system involves a single Glo3 enzyme recently discovered.

**Figure 2 antioxidants-11-02131-f002:**
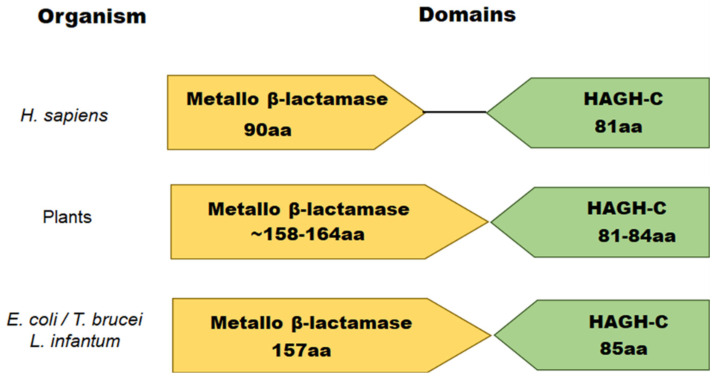
Schematic representation of the domains found in living systems for Glo2. Glo2 features a metallo-β-lactamase domain required for the catalytic activity and a hydroxyacylglutathione hydrolase C-terminus (HAGH-C) that represents the substrate-binding site. The length of these domains varies among different species and is indicated below each domain.

**Figure 3 antioxidants-11-02131-f003:**
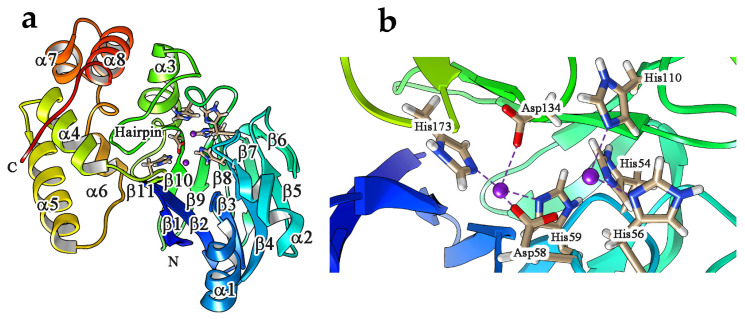
(**a**) Solid ribbon representation of human Glo2. The molecule has been color ramped according to residue number starting with red at the N terminus and finishing with blue at the C terminus. The metal ions and the coordinating residues are shown with balls and sticks. (**b**) Detail of zinc ion coordination by His and Asp residues at human Glo2 active site. The two images have been obtained from 1QH5 file.

**Figure 4 antioxidants-11-02131-f004:**
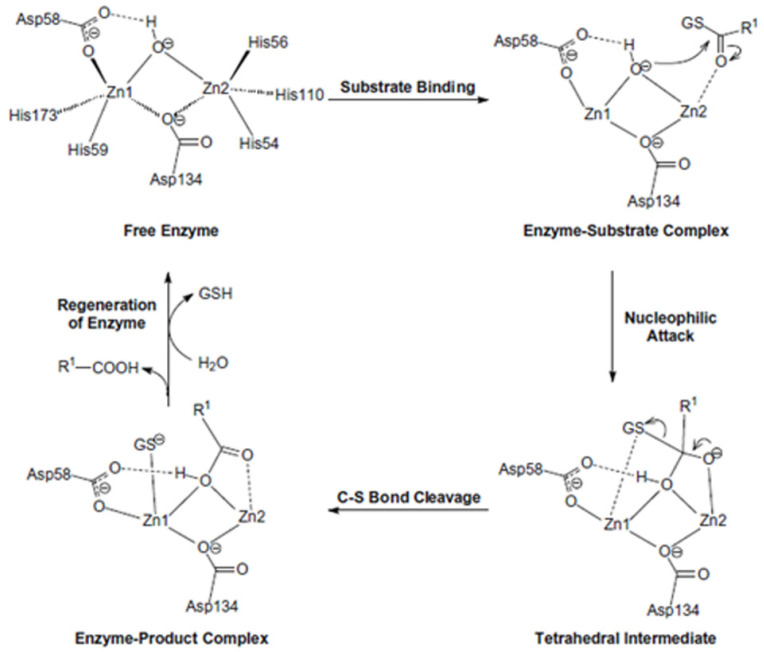
Proposed reaction mechanism for Glo2. Glo2 works via acid–base catalysis mediated by the nucleophile OH^−^ that can be generated under physiological conditions. After SLG binding (substrate binding), the nucleophilic attack of OH^-^ at the carbonyl carbon of lactoyl group of SLG takes place and a tetrahedral intermediate is formed (enzyme-substrate complex); Zn2 stabilizes this negatively charged intermediate. Next, the C–S bond of substrate breaks with help of the Zn1, resulting in a D-lactic acid and an unprotonated glutathione (GS^−^) (C–S bond cleavage). To complete the reaction, the GS^−^ group is protonated by a water molecule and the products are released from the active site to regenerate the enzyme (regeneration of enzyme). Reprinted from Journal of Inorganic Biochemistry, 103, Chen, S.L.; Fang, W.H.; Himo, F., Reaction mechanism of the binuclear zinc enzyme glyoxalase II—A theoretical study, 274–281, Copyright (2009), with permission from Elsevier.

**Figure 5 antioxidants-11-02131-f005:**
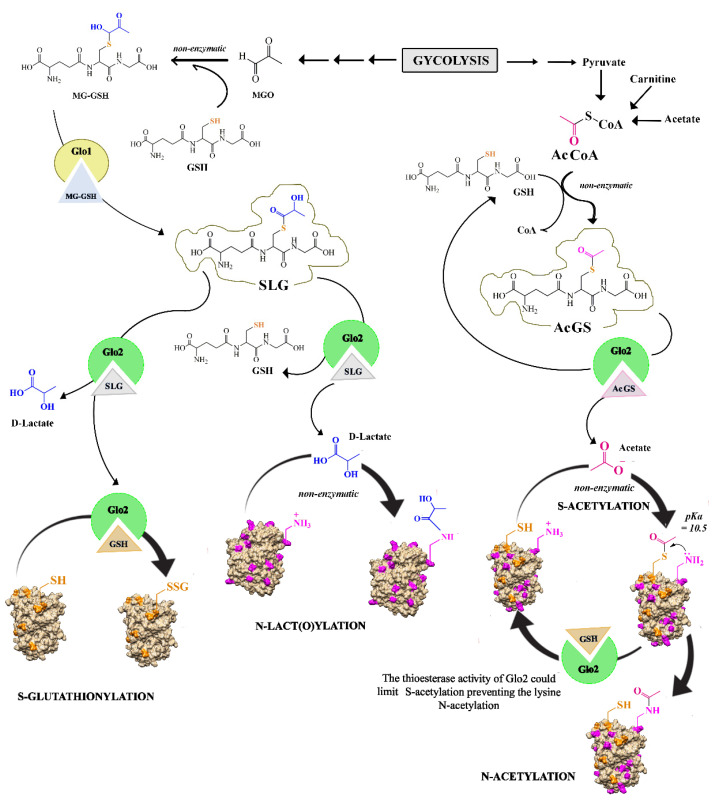
PTMs of lysine and cysteine residues involving Glo2. Glo1 converts 2-hydroxyacylglutathione (MG-GSH) into the thioester product *S*-D-lactoylglutathione (SLG). In *S*-glutathionylation, Glo2 catalyzes the hydrolysis of SLG, releasing D-lactate and transferring GSH in the GS^−^* form from its active site to the cysteine residues of redox-sensitive target proteins. In *N*-lact(o)ylation, the acyl group of the D-lactate is transferred non-enzymatically to the lysine residues of target proteins, generating a lactoyl-lys modification. Acetyl coenzyme A (AcCoA) and *S*-acetylglutathione (AcGS) reversibly acetylate protein cysteine residues (*S*-acetylation) followed by transfer of the acetyl moiety to a nearby lysine on mitochondrial proteins (*N*-acetylation). The thioesterase activity of Glo2 could limit *S*-acetylation and therefore the lysine *N*-acetylation.

**Figure 6 antioxidants-11-02131-f006:**
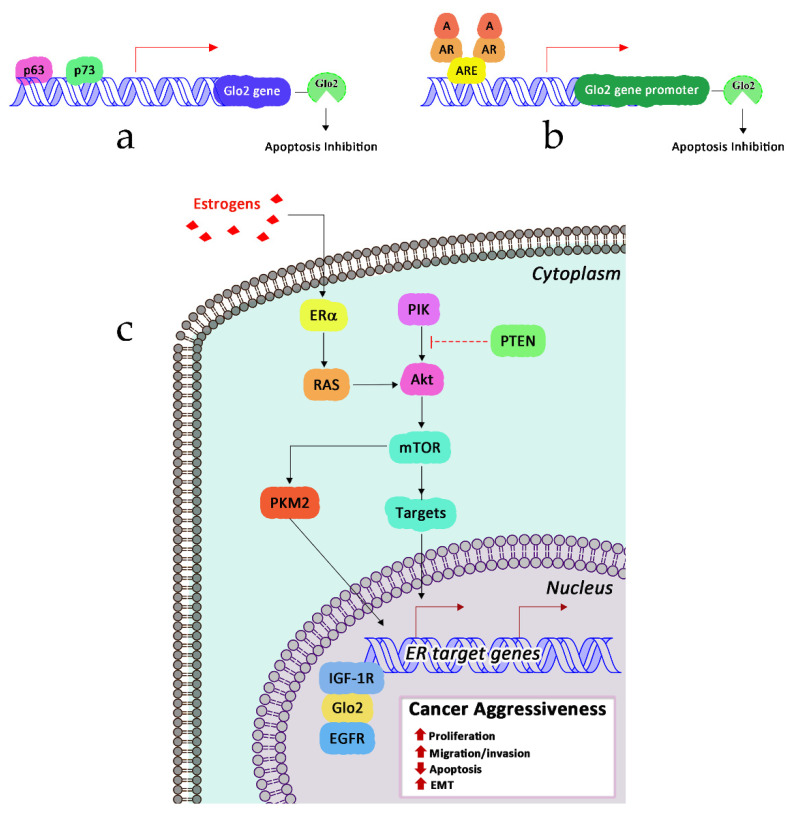
Glo2 and signaling pathways. (**a**) Transcription factor p63 and p73; (**b**) androgen receptor (AR); (**c**) phosphoinositide 3-kinase (PI3K)/protein kinase (AKT)/mammalian target of rapamycin (mTOR) signaling mediated by estrogen receptor alpha (ERα) activation acts as a positive regulator of Glo2 transcription. A, androgen; AR, androgen receptor; ARE, androgen response element; EGFR, epidermal growth factor receptor; IGF-1, insulin-like growth factor.

**Figure 7 antioxidants-11-02131-f007:**
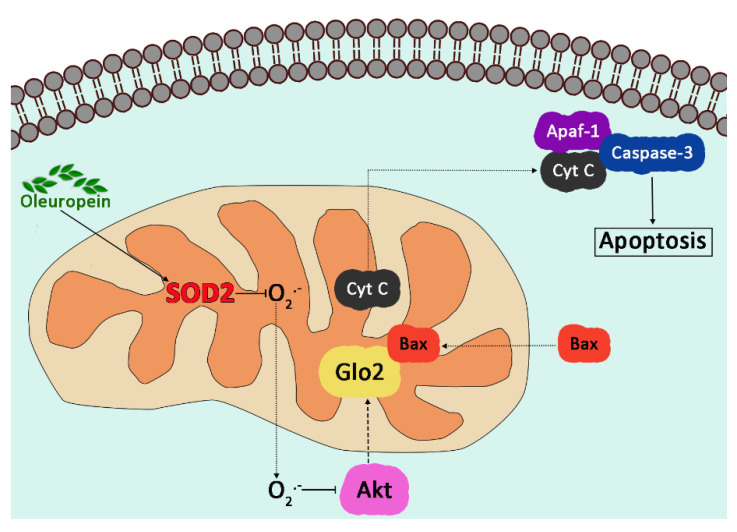
Oleuropein-induced apoptosis through mitochondrial Glo2.

## Data Availability

Not applicable.

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
