# Peer review of "Glyoxalase 2: Towards a Broader View of the Second Player of the Glyoxalase System"

_antioxidants, 2022, doi:10.3390/antiox11112131_

Round 1

Reviewer 1 Report

SUMMARY

I understand this to be an invited review. It is an enthusiastic attempt to cover the topic. I want to encourage the authors in their interest in Glo2. I suggest the authors take a critical look at their manuscript and try to improve the clarity and accuracy of the description. Some critical appraisal would also strengthen their contribution.

DETAILED REMARKS

Lines 34 – 36

Comment: Of course, Racker did not characterize the glyoxalase system well in the cited paper of 1951 because it was not recognized at this time the terminal product was D-lactate  - a vital aspect overlooked. It was rather suggested that the terminal product of methylglyoxal metabolism by the glyoxalase systems was D-lactate and not L-lactate of mainstream glycolysis in 1954, which was later confirmed by Mannervik and co-workers (E. Racker, Glutathione as coenzyme in intermediary metabolism. In Glutathione. (Ed. S. Colowick, A. Lazarow, E. Racker, D. R. Schwarz, E. Stadtman and H. Waelsch) p. 208, Academic Press, New York 1954; K. Ekwall, B. Mannervik. The stereochemical configuration of the lactoyl group of S-lactoylglutathione formed by the action of glyoxalase I from porcine erythrocytes and yeast. Biochim Biophys Acta 1973; 297: 297-9). For coverage of the historical development of glyoxalase research, please see [1],

Line 40

Comment: It should state “dihydroxyacetonephosphate” – not “hydroxyacetonephosphate”.

Lines 46 – 47

It is not correct to state that Glyoxalase 2 is the rate-limiting step of the glyoxalase system. Both reactions of the glyoxalase pathway, steps catalyzed by glyoxalase 1 (Glo1) and glyoxalase 2, function in situ in the steady-state such that the in situ concentrations of substrates are determined by flux through the pathway, enzyme concentration and concentration of GSH for Glo1. Both steps are under forward kinetic control; Glo1 is not at thermodynamic equilibrium in situ – as would be the case if Glo2 was the rate-limiting step. I am aware it has been stated in the literature in the past that Glo2 is the rate-limiting step but understanding of the glyoxalase pathway has advanced since then and the statement of the past on this is now known to be incorrect.

Further sections.

Comment: I appreciate the authors are attempting to cover all of the literature on S-D-lactoylglutathione (SLG). The comments appear to be description of published work without critical insights and appraisal. Perhaps the authors have not tried to measure cellular levels of SLG or model concentrations of it mathematically. Measurement of cellular concentrations of SLG by the highly sensitive and specific technique of stable isotopic dilution analysis LC-MS/MS have produced estimates of <0.4 µM of <0.02% total glutathione and mathematical models predict the cellular concentration is likely to be 1 – 2 nM [2, 3]. From the work cited herein, 1 mM SLG produces protein lactoylation. This is therefore an extremely supraphysiological concentration of SLG (likely 1,000,000-fold higher than present in cells physiologically) and provides little insight into the function of SLG physiologically in situ. From the very low level of SLG in cells it can be reasonably deduced: (i) it is highly unlikely to be a substrate for GSH uptake into mitochondria – see [2]; and Glo2 is not involved in regulation of protein lactoylation but rather fulfilling a cytoprotective function maintaining very low, tolerable levels of SLG to prevent protein lactoylation; cf. catalase has a cytoprotective function preventing protein oxidation by hydrogen peroxide – not regulating protein oxidation.

I suggest the authors read through their manuscript and reflect to see if critical appraisal and further insights can be made.

REFERENCES

1.         Seminars in Cell and Developmental Biology 2011; 22: 293-301

2.         Biochem. Soc. Trans 2014; 42: 419–424

3.          BMJ Open Diabetes Research & Care 2020; 8: e001458

Reviewer 2 Report

This review is about the glyoxalase 2 as second enzyme of the well known glyoxalase system, which is widely distributed from bacteria to men. This system is involved in detoxification of the toxic electrophile methylglyoxal (MG), which is produced as byproduct during the glycolysis and also from amino acid and sugar oxidation, resulting in protein modifications and advanced glycation end products. The review focuses on the genetics, molecular and structural properties, post-translational modifications, metabolic interactions and the role of diseases of glyoxalase-II (Glo2). A review solely focused on Glo2 has not been published, which was justified as reason of this review, since there is otherwise a wealth of literature reviews about MG detoxification using glyoxalases.

Overall, this review is well written, very comprehensive and provides especially interesting information about the roles of Glo2 in the different post-translational modifications, they authors have described, such as S-glutathionylation, S-acetylation and N-lactylation. The review is well structured, but the figures and legends are partly very brief and not very detailed and must be improved. In addition, the English Grammar and spelling must be improved due to numerous errors and incorrect sentence structures. I recommend to correct and proofread the English grammar and spelling together with a native speaker.

The following textual corrections are required to improve the manuscript:

Line 18: What means “located on 16p13.3” ? Could you please clarify ?

There is some redundancy and repetition in the abstract, which must be corrected.

Line 19-23: Correct as follows:

“This enzyme is the second enzymes of the glyoxalase system that is responsible for detoxification of the toxic electrophile methylglyoxal (MGO) in cells. The two enzymes Glyoxalase 1 (Glo1) and Glyoxalase 2 (Glo2) form the complete glyoxalase pathways, which depends on glutathione as cofactor in eukaryotic cells. The importance of Glo2 is highlighted by its ubiquitous distribution in prokaryotic and eukaryotic organisms.”

Line 27-28: This last sentence is unusual, it does not provide useful information and should be deleted:

“Since a review focusing exclusively on Glo2 has never been published, we hope that this work will provide a useful reference for researchers.”

Line 34:  Correct to: “The mechanism of action of the glyoxalase pathway was first well characterized in 1951 by Racker (Referenz). He demonstrated that …” (separate into two sentences)

Line  36-37: “conversion mechanism” is not a good description for the catalytic reaction. The exact catalytic functions of both glyoxalases have been described first.

Line  40: Correct: “dihydroxyacetonphosphate (DHAP)  and glyceraldehyde 3-phosphate, “

Line  41: Correct: “MGO is an alpha-ketoaldehyde”

Line  42-43: High concentrations of MGO modify proteins and nucleic acids, forming advanced glycation end-products (AGEs).- What are the specific modifications of the amino acids and which part of the nucleic acid is modified ? The detailed modification has not been explained and must be added.

Line  47-49: “it is proposable that Glo1 plays a role in the quick MGO detoxification, while the regulation of cellular redox state and metabolism depends instead on Glo2 activity.”

What are the underlying data supporting this hypothesis ? Glo1 cannot detoxify MGO alone. This thesis seems highly speculative in the present state.

Line 55: Correct this sentence as follows: “In various human parasites, the glyoxalase system has been studied as a potential drug target and especially as antimalarial target, given the importance of MGO as toxic electrophile, which has to be removed.”  

Line 62: This sentence is confusing and needs to be corrected and shortened to:

“The active site contains a binuclear metal center with Zn2+ as metal ion.”

Line 69: The abbreviation of S-D-lactoylglutathione is SLG (not LSG) according to the previous publications by the group https://doi.org/10.1016/j.freeradbiomed.2013.12.005. Please correct SLG throughout. Also S-glutathionylation and S-D-lactoylglutathione should have the “S-” in italics, please correct throughout.

Line 70: Correct: S-glutathionylation is the specific post-translational modification (PTM) of protein cysteine residues, which are oxidized to mixed protein disulfides with the tripeptide glutathione.

Line 73: Correct:…N-acetylation of the ε-amino group of lysine residues… “N-” should be in italics

Line 76-79: This sentence is hard to understand without further details and needs to be rephrased and explained in more detail and separate in two sentences at least:

“It has been shown that non-enzymatic acetylation of lysine residues in mitochondrial proteins often occurs through reversible S-acetylation of a proximal thiol and Glo2, degrading S-acetylglutathione, shifts the equilibrium away from cysteine S-acetylation, thus limiting acetylation of vital cysteine and lysine residues”

Please rephrase and clarify how S-acetylglutathione is formed and which reactions occur exactly to the final lysine acetylation. Also for better understanding the biochemical reactions for the different PTMs should be clarified in a figure as well, including S-acetylation, lysine acetylation  and lysine lactoylation.

Line 84: “lysines” (not lysins)

Line 85-86: Correct to: “In cancer cells, it has been shown that the upregulation of Glo2 with a proposed p63/p73-Glo2 regulatory axis could play a tumour-promoting role, at least in some cancer types.”

Line 100: Correct to: “S-lactoylglutathione lyase” as Glo1.

Line 103-105: Correct to: “The system is aimed to convert α-oxoaldehydes into the corresponding α-hydroxyacids quickly and efficiently. Typically, the major physiological α-oxoaldehyde removed is methylglyoxal that is converted to D-lactic acid via the SLG intermediate.” (Two sentences, SLG)

Line 108: Correct: isomerization to SLG

Line 108-109: Correct to: “In the second reaction, Glo2 hydrolyses SLG to the final product D-lactic acid and simultaneously regenerates the GSH consumed in the first reaction.”

Line 115: Correct to: “detoxification of MGO”

Figure 1 legend: Please add the enzymatic function of Glo1 and Glo2 in the legend for description.

Line 125: Correct to: “in the rice genome”

Line 134: Correct to: “Some genes encode inactive Glo2 forms, which are found for one of the three rice Glo2 and two of the five Arabidopsis Glo2.” Also what are the functions of inactive Glo2 forms and why are they inactive ? This should be shortly explained.

Line 138: Correct to: “but is not necessary in normal growing condition”

Line139: Correct to: “conditions”

Line141-142: Correct to: “The accumulation of MGO resulted in inhibition of germination and cell proliferation in a dose-dependent manner.”

Line148: Correct to: “Overexpression of Glo2 in rice and tobacco” (The protein is overexpressed, not the gene)

Line163-164: Correct to: “In addition to abiotic stressors, glyoxalase genes are highly induced by biotic stress conditions in bacteria, fungi, viruses, parasites, and insects.” I assume the authors mean that the glyoxalase are induced under the stress conditions, is that correct ? Please clarify

Line170: Correct to: “The regulatory mechanisms for the glyoxalase expression remains unclear…”

Line173: Correct to: “Glo2p and Glo4p proteins”

Line179: Correct to: “Analyses with mutants lacking either one or both glyoxalase 2 genes showed that..”

Line182: Correct to: “iii) to obtain an active Glo4p protein in E. coli through heterologous expression, the putative mitochondrial transit peptide at the N-terminus had to be removed”

Line187: Correct to: “The first report of a glyoxalase pathway”

Line189-190: Correct to: “Given the relevance in cellular detoxification of methylglyoxal, the glyoxalase system has been studied as potential drug target in some human parasites, such as Plasmodium falciparum, Leishmania spp. and Trypanosoma spp.”

Line191: Correct to: “Difference in the glyoxalase pathways”

Line193: Correct to: “encoding Glo2 enzymes”

Line200-202: Correct to: “This excessive glucose consumption allows rapid endomitotic nuclear divisions and drastic increase of parasitemia [65]. As consequence of this high glucose metabolism, both malaria parasites and their host cells show elevated MGO production [60]. (Two sentences)

Line204-206: Correct to: “In contrast to most eukaryotic organisms, the glyoxalase system of the trypanosomatid, including Leishmania spp. and Trypanosoma spp., uses reduced trypanothione (N1,N8-bis(glutathi-206 onyl)spermidine) (TSH), an alternative low molecular weight (LMW) thiol, as the preferred substrate.”

Line207-209: This sentence is hard to understand, please rephrase and clarify the functions of trypanothione (TSH) and trypanothione reductases (TR) in the parasites.

Line211: Correct to: “enzymes involved in the trypanothione biosynthesis pathway”

Line215: Correct to: “GSH” (instead of GHS)

Line222-224: Correct to:

“Gram-positive bacteria include bacilli (e.g., Bacillus subtilis, Bacillus anthracis, Bacillus cereus, Bacillus megaterium, Bacillus pumilis), staphylococci (e.g., Staphylococcus aureus, Staphylococcus saprophyticus) and streptococci (Streptococcus agalactiae), which produce bacillithiol (BSH) as alternative LMW thiol, which serves similar metabolic and redox cofactor functions as shown for GSH in eukaryotes.

(Note, abbreviation for Bacillus and Staphylococcus only after the species have first introduced)

Line225: References [75-78] are not related to bacillithiol, instead to mycothiol.

Please cite here instead the actual review about the synthesis and function of bacillithiol and the article about the identification of BSH in firmicutes:

-Chandrangsu P, Loi VV, Antelmann H, Helmann JD. The Role of Bacillithiol in Gram-Positive Firmicutes. Antioxid Redox Signal. 2018 Feb 20;28(6):445-462. doi: 10.1089/ars.2017.7057. Epub 2017 Apr 24. PMID: 28301954; PMCID: PMC5790435.

-Newton GL, Rawat M, La Clair JJ, Jothivasan VK, Budiarto T, Hamilton CJ, Claiborne A, Helmann JD, Fahey RC. Bacillithiol is an antioxidant thiol produced in Bacilli. Nat Chem Biol. 2009 Sep;5(9):625-7. doi: 10.1038/nchembio.189. Epub 2009 Jul 5. PMID: 19578333; PMCID: PMC3510479.

Line225-228: Correct to: “Recently, some researchers have started to investigate the role of BSH in the detoxification of reactive oxidants and electrophiles, such as methylglyoxal [79] and fosfomycin [80] as well as the protection and redox regulation of protein functions by protein-S-bacillithiolation during oxidative stress [81,82].

The review above (Chandrangsu et al., 2018) should be also cited here, which summarizes all functions of BSH in several firmicutes.

Line229-230:

This sentence must be deleted, since the role of BSH and glyoxalases has been studied in B. subtilis cells. The authors should refer to this paper and cite the obtained results about GlxA/B function in BSH-dependent detoxification of MGO in B. subtilis:

-Chandrangsu P, Dusi R, Hamilton CJ, Helmann JD. Methylglyoxal resistance in Bacillus subtilis: contributions of bacillithiol-dependent and independent pathways. Mol Microbiol. 2014 Feb;91(4):706-15. doi: 10.1111/mmi.12489. Epub 2014 Jan 7. PMID: 24330391; PMCID: PMC3945468.

Line230: Correct to: “In vertebrates, the single gene encoding glo2 produces two transcripts of 9 and 10 exons respectively.”

Line237/238: Correct to: “The molecular mass of the cytosolic form of Glo2 is around 29 kDa while that of the mitochondrial is in the range of 34 kDa.”

Line 238: Correct to: “The different molecular masses of the two isoforms are 238 due to ”

Line 239-241: Correct to: “This amino-terminal extension contains a mitochondrial targeting sequence, since it has been shown by confocal fluorescence microscopy that the mitochondrial form of Glo2 is directed to the..”

Line244: Correct to: “in the matrix and the intermembrane space”

Line246: Correct to: Arabidopsis in italics

Line250: Correct to: “SLG”

Line251: Correct to: “by limiting”

Figure 2: The legend is very brief and could be extended to two domains at least. Correct: T. brucei, E. coli. Also the figure is not very informative showing only the two domains. Given the large body of information about the different isoforms of Glo2, there could be a more detailed Table or Figure summarizing all the Glo2 forms and species, which are mentioned in the section.

Line275 and before: include the charges of the metal ions throughout as superscripts

e.g. “can bind Zn2+, Fe2+, and Mn2+

Line281: Correct to: “The enzymatic activity was found salt and pH sensitive.”

Line282: Correct to: “decreases with high NaCl concentration”

Line288: “The function of Glo3 could be also mentioned in the introduction and its biochemical reaction included in Figure 1.

Line292: Correct to: “only few studies reported different results on human, A. thaliana and P. falciparum Glo2 [90,93,94].”

Line304: Correct to: “308 amino acids”

Line317: Correct legend of Fig. 3 to: “are shown with balls and sticks”

Figure 4. The legend is again to brief. Please shortly summarize each step of the catalytic reaction.

Line344: Correct to: “Zn2 only stabilizes the tetrahedral intermediate. This role that can be also assumed by other metals such as Fe(II) or Ni(II) in Glo2 homologs of other species”

Line352-355: The sentences need to be rephrasend and clarified. Correct to:

“It is interesting to note that for Salmonella Typhimurium [109], a metal-selective product inhibition was observed, indicating that the iron variant is able to form a stable enzyme-product complex, while the manganese derivatives is not. The different metal forms of Glo2 during Salmonella infection could be exploited as a mechanism to regulate the enzyme activity.”

Line364: Correct to: “it was shown that”

Figure 5. The legend is again to brief. Please shortly summarize the steps leading to the PTMs.

Line372: Correct to: “SLG” see above

Line 399-400: This sentence is hard to understand as written. Correct to:

“The specificity of Glo2 towards actin is very interesting because actin is fundamental for the cell shape and function by playing mechanical, organisational and signalling roles.”

Line401-402: Correct to: “Reversible S-glutathionylation regulates actin polymerization showing an inhibition when Cys374 is S-glutathionylated.”

Line 411: Correct to: “Dalle Donne et al showed in vitro the S-glutathionylation of actin, which was achieved with a high concentration of non-physiological GSSG [117,118] and showed that...”

Line414: Correct to: “showed”

Line455: Correct to: “James et al identified”

Line456: Correct to: “these authors showed”

Line467: Correct to: “lysines"

Round 2

Reviewer 2 Report

Please see my comments for corrections of typos and errors marked in the pdf version of the manuscript using the comment tool.

The authors need to carefully proofread and correct again the manuscript.

-Species names, such as Arabidopsis thaliana must be written with full name only once and then abbreviated as A. thaliana. This has to be corrected for all species throughout and is often incorrect.

-in vitro, in vivo must be in italics and corrected throughout

-glyoxalase 1 and glyoxalase 2 must be written with small letters, not large letters
